# Epidemiology and Comparative Analyses of the S Gene on Feline Coronavirus in Central China

**DOI:** 10.3390/pathogens11040460

**Published:** 2022-04-12

**Authors:** Hehao Ouyang, Jiahao Liu, Yiya Yin, Shengbo Cao, Rui Yan, Yi Ren, Dengyuan Zhou, Qiuyan Li, Junyi Li, Xueyu Liao, Wanfeng Ji, Bingjie Du, Youhui Si, Changmin Hu

**Affiliations:** 1State Key Laboratory of Agricultural Microbiology, Huazhong Agricultural University, Wuhan 430070, China; ouyanghh@webmail.hzau.edu.cn (H.O.); sbcao@mail.hzau.edu.cn (S.C.); zhoudy6@webmail.hzau.edu.cn (D.Z.); lqylqy6@webmail.hzau.edu.cn (Q.L.); dubingjie@webmail.hzau.edu.cn (B.D.); 2College of Veterinary Medicine, Huazhong Agricultural University, Wuhan 430070, China; jiahaoliu@webmail.hzau.edu.cn (J.L.); yinyiya@rp-pet.cn (Y.Y.); yanrui@webmail.hzau.edu.cn (R.Y.); renyi0818@webmail.hzau.edu.cn (Y.R.); lijunyi@webmail.hzau.edu.cn (J.L.); liaoxueyu@webmail.hzau.edu.cn (X.L.); wanfeng0609@webmail.hzau.edu.cn (W.J.); 3Hubei Hongshan Laboratory, Wuhan 430070, China

**Keywords:** FCoV, S gene, mutation, seroprevalence, hydrophobicity

## Abstract

Feline coronavirus (FCoV) infections present as one of two forms: a mild or symptom-less enteric infection (FEC) and a fatal systemic disease termed feline infectious peritonitis (FIP). The lack of epidemiology of FCoV in central China and the reason why different symptoms are caused by viruses of the same serotype have motivated this investigation. Clinical data of 81 suspected FIP cases, 116 diarrhea cases and 174 healthy cases were collected from veterinary hospitals using body cavity effusion or fecal samples. Risk factors, sequence comparison and phylogenetic studies were performed. The results indicated that FIPV was distinguished from FECV in the average hydrophobicity of amino acids among the cleavage sites of furin, as well as the mutation sites 23,531 and 23,537. FIPV included a higher minimal R-X-X-R recognition motif of furin (41.94%) than did FECV (9.1%). The serotype of FCoV was insignificantly correlated with FIP, and the clade 1 and clade 2 strains that appeared were unique to central China. Thus, it is hypothesized that this, along with the latent variables of an antigenic epitope at positions 1058 and 1060, as well as mutations at the S1/S2 sites, are important factors affecting FCoV transmission and pathogenicity.

## 1. Introduction

Ever since the outbreak of COVID-19, great attention has been given to coronavirus, its epidemiology, prevention and diagnosis [1]. FCoV, one of the best-known coronaviruses in veterinary medicine, may be of great interest to researchers [2]. Some surveys of the prevalence of feline coronavirus demonstrated that FCoV colonization of the alimentary tract is prevalent in most feline populations, with carriage rates exceeding 20% in northern China (80.0%), southern China (73.1%), southwestern China (80.35%), eastern China (74.6%), Portugal (47.5%), Germany (76.5%), Malaysia (84%), southern Italy (80%) and Japan (37%) [3,4,5,6,7,8]. However, there is currently a lack of epidemiological statistics on the disease in central China.

Feline coronavirus (FCoV) can be divided into two serotypes: type Ⅰ and type II [9,10,11]. Serotype Ⅰ is the most primitive feline coronavirus, and serotype II is a recombination of feline coronavirus type I and canine coronavirus [11,12]. Additionally, it can also be divided into two biotypes: FECV and FIPV. FECV is highly contagious, as shown by Pedersen et al. [13]. The clinical appearance of FECV, if any, is characterized by mild enteritis [14]. Comparatively, FIP was first reported in the literature in 1966 and is caused by FIPV, a systemic, highly lethal feline infectious disease that spreads worldwide [15]. It was reported that 75.7% of suspected FIP was coronavirus-positive [4]. It is probable that mutations in some sites, such as ORF 3c, the spike (S) gene and ORF 7b of FECV, would change the enterocyte tropism to monocyte/macrophage cells, and especially change the symptoms from mild enteritis to a fatal symptomatic disease [16].

FCoV is an enveloped, unsegmented, single-stranded positive-sense RNA virus (ssRNA+), which belongs to Nidovirales (order), Coronaviridae (family), Alphacoronavirus (genus) and alphacoronavirus 1 (species) in viral taxonomy. The genome length is approximately 28–30 kb, which contains 11 open reading frames (ORFs) encoding a polyprotein with RNA synthesis function (1a, 1b), four structural proteins (highly glycosylated spike protein (S), envelope protein (E), membrane protein (M) and nucleocapsid protein (N)) and several non-structural accessory proteins (3a, 3b, 3c, 7a, 7b). The UTR region at the 3′ end of the FCoV genome is a conservative untranslated region [17,18,19,20]. Earlier studies showed that mutations at positions 23,531 and 23,537 on the S gene could increase the probability of FIP [21]. In contrast, some scholars reported that mutations at these two sites change the tissue tropism of the virus but are uncorrelated with FIP [22]. To date, the precise cause of the switch from FECV to FIPV is still poorly understood.

For collection, suspected FIP cases need to combine clinical diagnosis, laboratory diagnosis and other aspects. The main clinical symptoms of FIP are depression, weight loss, loss of appetite and increased body temperature fluctuations. Most clinical cases are exudative FIP, with accumulation of fluid in the body cavity and serositis being significant pathological changes. Some nonexudative FIP manifests as extensive granuloma and vasculitis, as well as neurological and ocular symptoms [23,24]. The ratio of albumin to globulin can also be used as a standard for the clinical diagnosis of FIP [25]. In the present study, these values were used as reference standards for collected cases to analyze the differences in the S gene of coronaviruses carried by these cats with FIP.

Most of the known coronaviruses contain a furin cleavage site at the S protein. Relevant reports on human coronaviruses indicate that the deletion of this site significantly reduces the pathogenicity of human coronaviruses [26,27]. This suggests that amino acid changes at furin sites could have an impact on the pathogenicity of feline coronaviruses. However, it is still unclear which sites have changed and caused the impact. In 2013, Licitra et al. suggest that mutations at furin cleavage sites will change the efficiency of furin’s action on the peptide chain [28]. In 2014, Izidoro et al. reported that furin cleavage in the S1/S2 site was incompatible with a hydrophobic aliphatic side chain [29]. There seems to be a certain correlation between them, and its correlation with FIP still needs to be supported by clinical data.

In this study, a molecular epidemiological investigation, which was mainly focused on central China, was conducted for FCoV. The analysis of partial S gene genomes substantiated the genetic evolution of the identified FCoV strains and described the distinction of FECV and FIPV at the genetic and amino acid levels. The purpose of this research was to provide insight into the epidemiology and genetic diversity of the FCoV strains widespread in central China and to distinguish FECV and FIPV. Valuable information for the diagnosis, prevention and control strategies of FCoV infections will be provided by the experimental data.

## 2. Materials and Methods

### 2.1. Study Design

This article is a retrospective study incorporating molecular epidemiology. In order to clarify the difference between FIPV and FECV, the previously collected cases and their related information (clinical status, sex, breed, age and residential density) were analyzed. Combined with RNA extracted from the disease materials of the corresponding case for reverse transcription, the target fragment was amplified by PCR, and the sequence was obtained by sequencing. In order to confirm whether typing can distinguish FIPV from FCoV, we performed phylogenetic analysis of the sequences and obtained an evolutionary tree. Combined with the FIP performance, differential analysis and statistical analysis were performed on the sequences to distinguish the reason why different symptoms are caused by feline coronaviruses of the same serotype.

### 2.2. Sample Collection

From May 2018 to March 2021, 371 cases were collected from veterinary hospitals, among which 81 cats presented with symptoms of suspected FIP and 116 cats presented with symptoms of diarrhea. The remaining 174 were healthy cats. Standard virus sampling tubes were used for temporary storage of fecal samples. Both fecal samples and body cavity effusion were stored at −80 °C.

Both immunohistochemistry (IHC) and immunofluorescence are the gold standard for detecting FIP [30]. Since FIP has no specific diagnostic method and immunofluorescence staining diagnosis is not possible in most hospitals, the collection of FIP suspected cases in this study is based on IHC or by veterinarians using various methods (mainly including: ultrasonic, X-ray, hematological and biochemical analyses, RT-nPCR, inappetence, weight loss, lassitude, cachexia, dyspnea, ocular signs, neurological signs, icterus, fever, body cavity effusion, abdominal mass and diarrhea, etc.). After initial clinical diagnosis, suspected FIP cats were also diagnosed using necropsy (suppurative granuloma, membranous glomerulonephritis), histopathological examination and RT-PCR with primers designed for the highly conserved region (3′-UTR) of the FCoV test. Tissue samples with a positive IHC test in suspected FIP cases and a positive RT-PCR test in ascites were regarded as FIP confirmed cases, and the successfully sequenced samples were used as materials for subsequent molecular epidemiological analysis. Detailed information on the primers used in this study can be seen in Table 1.

### 2.3. RNA Extraction and Reverse Transcription

The extraction of RNA refers to the method of Geng et al. [31]. Briefly, after brief centrifugation to remove debris (3 min at 12,000× *g* at 4 °C), RNA was extracted with TRIzol. The HiScript^®^ II Q RT SuperMix for qPCR (+gDNA wiper) test kit was used, following the instructions to reverse transcribe RNA into cDNA. The cDNA product was used immediately or stored at −80 °C to avoid repeated freezing and thawing.

All the amplified products were sent to AuGCT Biotechnology Co., Ltd. (Wuhan, China) for sequencing.

### 2.4. The Detection of FCoV for 3′-UTR

The primers F1 and R1 were designed for the 3′-UTR, a highly conserved region of the FCoV genome. Most FCoV strains have been detected by this region. The PCR reaction volume was 25 µL. For PCR amplification, 2 × PCR Taq Plus MasterMix (Applied Biological Materials (abm) Inc., Vancouver, BC, Canada) was used. The expected size of the product was 223 bp.

### 2.5. FCoV Serotyping

To genotype the FCoV strains collected in our study, the 3′ end of the S gene was amplified by reverse transcription-nested polymerase chain reaction (RT-nPCR) using the primers designed by Lin et al. (2009). The first round of PCR primers was F2 and R2, assuming the product size was 702 bp. The nested primers were F3, F4 and R3 (F3 and F4 were forward primers designed specifically against the type I and II FCoV sequences, respectively; R3 was a universal reverse primer). The expected sizes of the type I and type II FCoV products were 360 bp and 218 bp, respectively. Two rounds of PCR products were verified by 1% agarose gel electrophoresis.

### 2.6. Phylogenic Analysis

The target DNA fragment was purified, and the nucleic acid sequence was determined by sequencing. The reference S gene of typical FCoV strains from China and other regions was retrieved from the National Center for Biotechnology Information (NCBI) nucleotide database. All the details about the nucleic acid sequence in this study were uploaded to GenBank, as shown in Appendix A. Additionally, the referenced strain-related gene bank number is provided in Appendix A. After sequence alignment by MEGA X (64-bit) software, the phylogenetic tree was constructed using the neighbor-joining method. The NJ phylogenetic tree was established by using the p-distance method with 1000 bootstrap replications. Then, the tree was organized and simplified by using the Interactive Tree of Life (iTOL) version 5 [32]. Additionally, to identify FCoV genotypes, three distinct methods were used, including MCC tree (BEAST v1.8.4) [33], ML tree (RAxML v8.2.12) [34] and NJ tree (MEGA X v10.0.2).

### 2.7. FCoV Mutation Site 23,531 and 23,537 Detection

As mentioned in a previous review, it is believed that there were two amino acid substitutions, M1058 L and/or S1060A, corresponding to nucleotide mutations in the S protein of FCoV identified by Chang et al. To verify whether the FCoV mutation detection of M1058 L and S1060A is suitable for the diagnosis of FIP, RT-nPCR was performed on the samples already identified as FCoV-positive according to the primers [21].

The primers for the first round of PCR were F5 and R5, and the genome of FCoV was amplified from 23,442 to 24,040. In addition, the nested primers were F6 and R6, and the genome of FCoV was amplified from 23,451 to 23,593. The expected sizes of the first- and second-round products were 598 bp and 142 bp, respectively.

Sequence alignment was then performed to observe whether there were mutations at 23,531 and 23,537 of the S gene and to explore whether the detections of M1058 L and S1060A mutations in FCoV were suitable for the diagnosis of FIP.

### 2.8. Detection of Furin Cleavage in the S1/S2 Site

It is acknowledged that furin cleavage in the S1/S2 site is incompatible with a hydrophobic aliphatic side chain [29]. To explore the distinction of mutation and the hydrophobicity between FIPV and FECV in the S1/S2 site, primers F7, R7, F8 and R8 were designed by Licitra et al. for the PCR amplification of this fragment [28].

All data on protein hydrophobicity were obtained from the Swiss Institute of Bioinformatics, and graphing for hydrophobicity of amino acids at S1/S2 cleavage sites and statistical analysis were performed with GraphPad Prism 8 software (GraphPad Software, 147 Inc., La Jolla, CA, USA).

Sequence logos were generated using Weblogo. The reference sequences of FECV and FIPV were obtained from NCBI, and then, to give all sequences the same coordinated alignment, Genious Primer was used to trim the ends and gaps of the alignment sequences.

## 3. Results

### 3.1. FCoV Detection Based on the 3′-UTR

A total of 371 clinical samples, including 81 FIP suspectable cases, 116 diarrhea cases and 174 healthy cases, were collected from veterinary hospitals using body cavity effusion or their fecal samples. Feline coronavirus infection was detected by RT-PCR targeting the 3′-UTR of the S gene. The numbers (N) contributing to individual subscales vary slightly, since some questions were not answered by pet owners. Among the total clinical samples (n = 371), 173 cases were positive (46.6%, 173/371). Under classification by clinical presentation (n = 369), the positive rate of FIP-suspected cats reached 59.3% (48/81), while the positive rate of healthy cats was 42.8% (124/290). Divided by sex, the positive rate for male cats was 48.4% (90/186) and that for females was 44.2% (76/172). Roughly divided by breed, purebred cats had a positive rate of 51.8% (116/224), while that of mongrel cats (n = 127) was 36.2% (46/127). The positive rate in cats younger than 10 months (n = 211) was 51.7% (109/211) and that in cats older than 10 months was 38.7% (58/150). In terms of residential density, the positive rate of cats reared in groups was 58.1% (61/105), while that of cats reared alone was 37.2% (81/218) (Appendix A). Above all, it is significant that the prevalence of FCoV was associated with clinical status (*p* < 0.01), breed (*p* < 0.01), age (*p* < 0.05) and residential density (*p* < 0.01) but insignificantly correlated with sex.

### 3.2. FCoV Serotyping

To further investigate the evolution of FCoV in central China, the 3′ end of the S gene was amplified by RT-nPCR to genotype FCoV strains. As shown in Table 2, the FCoV-positive rate based on detection of the 3′ end of the S gene was significantly lower than that of 3′-UTR detection because of the reduced sensitivity. Of the 172 FCoV-positive samples, 85 S genes were successfully sequenced. The genotyping results indicated that all 85 samples were positive for type I FCoV (100%), and no coinfection with types I and II or type II FCoV infection alone was observed. Information related to these sequences in GenBank is provided in Appendix A.

Within the 85 sequenced samples, the numbers of FIP-suspected and non-FIP cats that tested positive based on the partial S gene for type I FCoV were 18 and 67, respectively. When almost all reported strains were added to the statistics in Table 3 (*p* = 0.648 > 0.05), it was found that FIP was insignificantly correlated with the serotype of the feline coronavirus. The referenced strain-related gene bank number is provided in Appendix A.

### 3.3. Phylogenic Analysis

To explore the genetic diversity and evolutionary relationship of feline coronaviruses, a phylogenetic tree was constructed using isolated feline coronavirus strains from central China (strains marked with red dots were detected in this study) and around the world. As shown in Figure 1, of the 85 FCoV strains, 85 belonged to the type I FCoV cluster and formed 17 clades in the phylogenetic tree based on the 3′ end portion of the S gene. None belonged to the type II cluster closely related to canine coronavirus and transmissible gastroenteritis coronavirus.

### 3.4. FCoV Mutation Site 23,531 and 23,537 Detection

To further explore the difference between FIPV and FECV, the detected measure reported by Chang et al. was applied. The 138 sequences (75 in this study and 63 from GenBank) related to mutation sites 23,531 and 23,537 have been well analyzed (Figure 2). The mutation of the A base at 23,531 to the T base results in the substitution of the amino acid M at position 1058 with the amino acid L. Additionally, mutation of the T base at 23,537 to a G base results in the substitution of amino acid S by amino acid A at position 1060.

There were 11 FIPV-related sequences whose A base at position 23,531 was mutated to a T base and two FIPV-related sequences whose T base at position 23,537 was mutated to a G base. The mutation at this site is significantly associated with the development of FIP (*p* < 0.01; Table 4).

### 3.5. Key Restriction Site Detection of Furin Protein in the S1/S2 Region of FCoV

This study considered that mutations at other sites of the S gene might also convert FECV to FIPV, while the key restriction sites of furin protein in the S1/S2 region were also sequenced in this study (Figure 3). The average hydrophobicity of amino acids around the cutting site was calculated and plotted by the software (Figure 4).

According to amino acid sequence analysis, the proportion of P at position −6 in the FCoV sequence (15.2%) was significantly lower than that at position −6 (19.4%) in the FIPV sequence. The sequence at position −4 of the FCoV sequence is relatively conserved, consisting of F (3.0%) and R (97.0%). However, FIPV showed diversity at the −4 site (R: 71.0%; G: 12.9%; T: 9.7%; Q: 3.2%; S: 3.2%). At position −3, the proportion of hydrophobic amino acid A in the amino acid sequence of FIPV (29%) is much higher than that of FCoV (24.2%). At position 2, the FCoV amino acid sequences are composed of T (75.7%), I (6.1%), A (6.1%), E (6.1%), P (3.0%) and E (3.0%), while the sequences of FIPV consist of hydrophobic amino acids A (19.4%), P (14.3%), V (6.5%) and E (3.2%) and hydrophilic amino acids S (3.2%) and T (28.6%). Through comparison, it was found that in the abovementioned positions, the ratio of hydrophobic amino acids in the sequence composition of FIPV was significantly higher than that of FCoV.

By sequence alignment, 90.9% of the amino sequences of FCoV contain the canonical R-X-K/R-R recognition motif, and 9.1% of the amino sequences of FCoV contain the minimal R-X-X-R recognition motif. A total of 54.84% of the amino sequences of FIPV contain the canonical R-X-K/R-R recognition motif. A total of 41.94% of the amino sequences of FIPV contain the minimal R-X-X-R recognition motif. A total of 3.23% of the amino sequences of FIPV do not include the furin cleavage site. The FIPV ratio with the minimal R-X-X-R recognition motif was significantly higher than that of FCoV.

A line graph of the average hydrophobicity change at these sites is presented in Figure 4. Point 0 in Figure 4 represents the furin cleavage site. This indicated that, for both the data statistics in this study and the sequence statistics after combining with the reference sequence, in the −5 to +5 region of the figure, the value of FIPV is significantly higher than that of FCoV, which is a specific numerical reflection of hydrophobicity.

## 4. Discussion

Earlier studies showed that mutations at positions 23,531 and 23,537 on the S gene could increase the probability of FIP. In contrast, some scholars reported that mutations at these two sites change the tissue tropism of the virus but are uncorrelated with FIP [21]. The precise cause of the switch from FECV to FIPV is still poorly understood. Aiming to deepen the understanding of FCoV epidemiology and the distinction between feline enteric coronavirus (FECV) and FIPV at the molecular level, this article provided analysis of known risk factors, a genetic evolutionary tree, mutations in sites 23,531, 23,537 and the cleavage sites of furin.

Focusing on known risk factors, this study augments the data on the prevalence of FCoV infection in central China. In this region, the FCoV positivity rate is 46.6%, which is significantly lower than that in northern China (80.0%), southern China (73.1%), Portugal (47.5%), southern Italy (80%), southwestern China (80.35%) and eastern China (74.6%) [3,4,5]. The FCoV positivity rate in cats with FIP was 59.3%, which is lower than that reported by Li et al. (75.7%) and higher than that reported in South Korea (19.3%) and Turkey (37.3%) [5,35,36]. Differences in data may be related to the number of samples, the way they were collected and stored and the specificity and sensitivity of the detection methods. The specific reasons for the low prevalence of feline coronavirus in the central region need to be further explored. It is speculated that it may be related to the feeding density (single cat or multiple cats) and breeding methods (domestic or free-range) of people in different regions.

Data analysis showed that the presence of FCoV was significantly associated with the presence of FIPV, indicating that FIPV likely evolved from FECV through mutations, and that the control of FCoV is very important in reducing the occurrence of FIP. In addition, the density of cats is significantly associated with the spread of FCoV. Currently, no vaccine prevents the development of FIP in China, where the fatality rate and treatment costs are high. Moreover, it was suggested that breeders could limit the density of cats to reduce the spread of FCoV and consequently reduce the incidence of FIP. Additionally, the data indicated that FCoV infection does not directly cause diarrhea (Appendix A) and that the vast majority of healthy cats similarly carry FCoV, which is an opportunistic pathogen that is likely to exhibit pathogenicity when its host experiences stress. The correlation between stress and FIP in cats has also been previously confirmed [25].

As observed in this study, all FCoV strains circulating in central China are statistically considered FCoV type I strains, and the occurrence of FIP does not seem to be associated with the serotype of the coronavirus (Table 3, *p* = 0.648 > 0.05). Overall, FCoV exhibits rich genetic diversity. As seen in the phylogenetic tree, FCoV is distantly related to the human coronavirus and is closely related to the canine and porcine coronaviruses. Similar to Herrewegh et al.’s opinion, FCoV type II might have originated from genetic recombination of canine coronavirus and cat coronavirus type I [11,12]. The type I FCoV clades 14, 16 and 17 might be unique to China, and clades 16 and 17 seem to be a branch of the FCoV that emerged in central China.

Chang et al. proposed that the base mutations at positions 23,531 and 23,537 of the S gene of FCoV have a greater effect on the evolution of FECV into FIPV. In 2014, Porter et al. found that the amino acid changes in the spike protein of FCoV are associated with the spread of the virus from the intestinal tract and that they are not associated with FIP [22]. In their research, 112 samples were collected, including 51 samples from 16 non-FIP cats; the sequencing results were obtained for nine out of 41 tissue samples and for six out of 10 stool samples. The results showed that the leucine codon was found in most (89%) of the non-FIP tissue samples, and a large number (9%) of FIP tissue samples had a methionine codon at this position. Therefore, it is concluded that the mutations at the two positions of the S protein have no association with the occurrence of FIP. The possibility of operation-related cross-contamination could not be considered given the small sample size.

In our research, leucine (5.6%) was found at the S1058 site in the samples obtained from the non-FIP cats. This finding indicated that the occurrence of a mutation at this position is indeed not a necessary and suitable condition for FECV to evolve into FIPV. Nonetheless, through statistical analysis, the relationships and conclusions postulated remain highly plausible, and it is reasonable that the mutation at this site is significantly associated with the development of FIP (*p* < 0.01; Table 4). Peter et al.’s idea that mutations at this locus increase the tissue orientation of FECV and render it possible for this virus to spread throughout the body is consistent with our conclusion. On this basis, the systemic transmission of FECV is associated with the development of FIP, and it does not rule out the possibility that mutations in other locations also cause the same effect.

The average hydrophobicity of the amino acids was calculated through sequencing and comparison (Figure 4). It was shown that the hydrophobicity of the FIPV sequence, which ranged from the −4 site to the +5 site, was significantly higher than that of the ordinary FCoV sequence. This finding is consistent with the results obtained based on the reference sequence data. Hydrophobic amino acids are not conducive to the action of furin.

Izidoro et al. reported that peptides bearing the RPP sequence are resistant to furin hydrolysis. In this study, it is believed that hydrophobic amino acids similar to P and A appear around the furin cleavage site, increasing the resistance of the peptide chain to furin, which might be one of the reasons why FECV evolved into FIPV. The internationally recognized typical FIPV strain FIPV-791146 and the typical strain FIPV/HN/2021/xina found in this study both lacked furin cleavage sites, also supporting the above proposition. Sequence comparison also revealed that nearly all FCoV type II strains do not contain furin cleavage sites. This lack of furin cleavage sites may also be the reason why type II FCoV is more likely to cause FIP in previous studies, and why the replicability of type I FCoV in feline cell lines is lower than that of type II FCoV in Felis catus whole fetus (fcwf)-4 cells [37].

## 5. Conclusions

FCoV spreads widely in cats in central China and is significantly related to the occurrence of FIP. The mutations at sites 23,531 and 23,537 of the S gene of FCoV are significantly related to the occurrence of FIP, and they can be used as a reference for clinical diagnosis. Mutations at S1/S2 sites are an important factor affecting FCoV transmission in vitro and pathogenicity in vivo. The replacement of hydrophobic amino acids at this site, the deletion of the cleavage site or the transformation of the virus into a difficult-to-recognize state are more likely to lead to FIP in the host.

## Figures and Tables

**Figure 1 pathogens-11-00460-f001:**
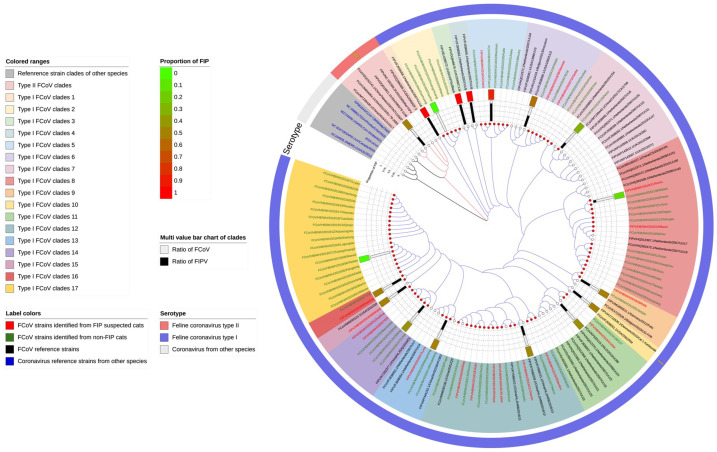
Phylogenetic analyses of FCoV strains on the basis of the partial S gene. The red dot diagram represents the 85 FCoV strains identified in our study. The number of FIP cats that tested positive based on part of the S gene for type I FCoV was 18 in this study. The number of FIP isolates obtained from GenBank for type I FCoV was 24. The number of FIP cats obtained from GenBank for type II FCoV was 3. The number of non-FIP cats that tested positive based on part of the S gene for type I FCoV was 67 in this study. The number of non-FIP cats obtained from GenBank for type I FCoV was 21. The number of non-FIP cats obtained from GenBank for type II FCoV was 3. The referenced strain-related gene bank number is provided in Appendix A. Color ranges: for the purpose of studying the genetic distance between different clades, the research has distinguished the different clades with different color ranges. In addition, reference strain clades of other species are marked in a gray color range. Label colors: the red font leaves represent the FCoV strains identified from FlP-suspected cats; the green font leaves represent the FCoV strains identified from non-FIP cats; the black font leaves represent the FCoV reference strains; the blue font leaves represent the coronavirus reference strains from other species. Serotype: in the outer circle of the phylogenetic tree lies the serotype of the coronavirus, which has been divided into three parts: feline coronavirus type II (red); feline coronavirus type I (purple) and coronaviruses from other species (gray). Proportion of FIP is presented as a histogram with black and white in each clade, above the bar value we show a green–red block to indicate the FIP prevalence of different clades. (Color figure can be viewed at https://raw.githubusercontent.com/lljjh/tree1/main/%E5%8F%91%E8%82%B2%E6%A0%91jpg.jpg.) (Accession date is after 1 January 2022).

**Figure 2 pathogens-11-00460-f002:**
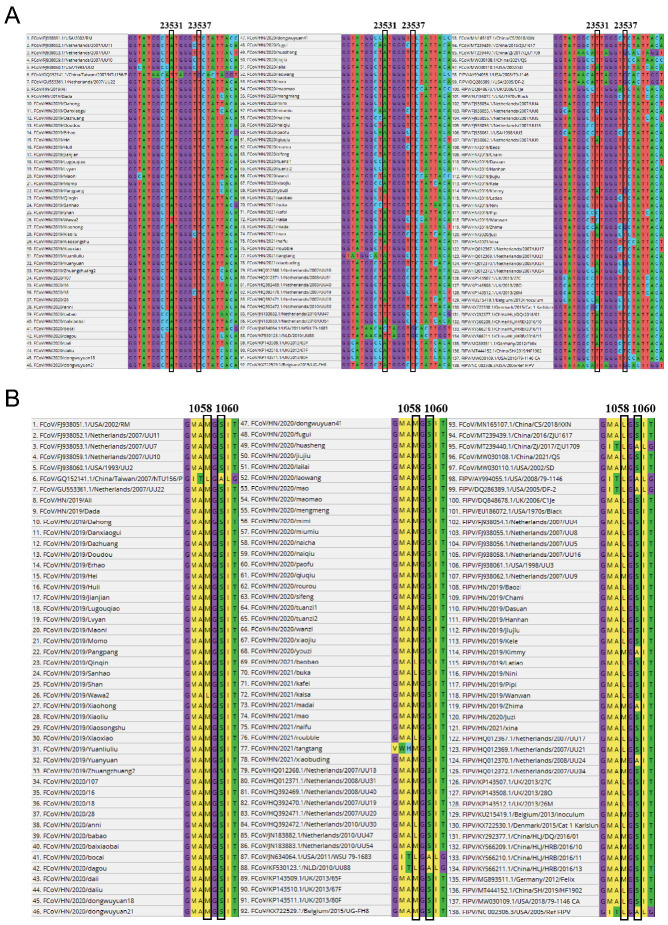
The mutation sites 23,531 and 23,537 of the detection results of nucleotide sequence translation of the fragments. (**A**) Alignment analyses of the nucleotide sequences of partial S genes between the identified FECV strains and FIPV strains. (**B**) Alignment analyses of the amino sequences of the corresponding partial S genes between the identified FECV strains and FIPV strains. The referenced strain-related gene bank number is provided in Appendix A. T: thymine; A: adenine.

**Figure 3 pathogens-11-00460-f003:**
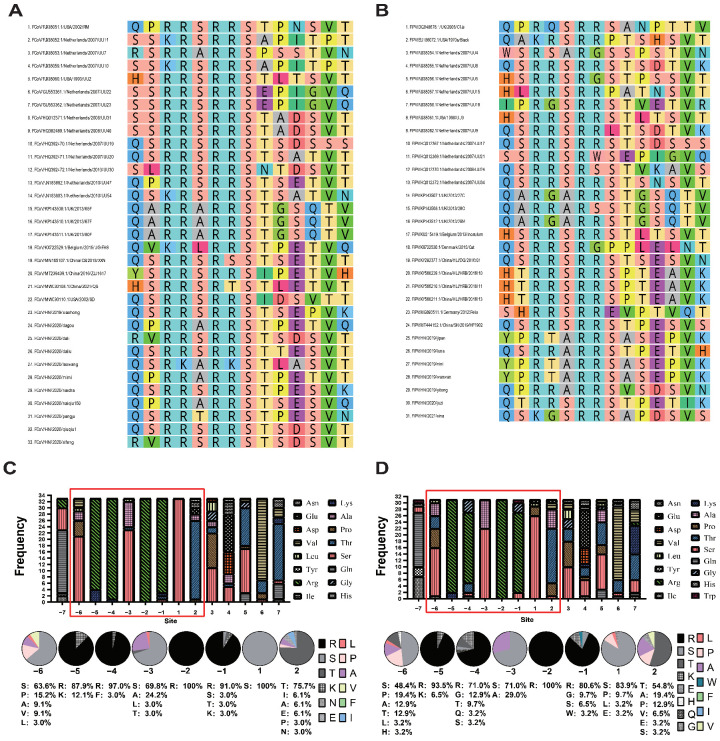
Sequence analysis of feline enteric coronavirus (FECV) and feline infectious peritonitis virus (FIPV) in the spike gene at the S1/S2 site. (**A**) Amino acid sequence alignment of feline enteric coronavirus (FECV) at the S1/S2 region of S proteins aligned by Geneious Primer. (**B**) Amino acid sequence alignment of feline infectious peritonitis virus (FIPV) at the S1/S2 region of S proteins aligned by Geneious Primer. (**C**) Sequence analysis of feline enteric coronavirus. The relative frequency with which an amino acid appears at the S1/S2 position is reflected by the color, as depicted by the scale bar. The pie chart represents the percentage of amino acid sequences in the S1/S2 region. (**D**) Sequence analysis of feline infectious peritonitis virus. The probability of amino acids is indicated by the size of the letters. The relative frequency with which an amino acid appears at the S1/S2 position is reflected by the color, as depicted by the scale bar. The pie chart represents the percentage of amino acid sequences in the S1/S2 region.

**Figure 4 pathogens-11-00460-f004:**
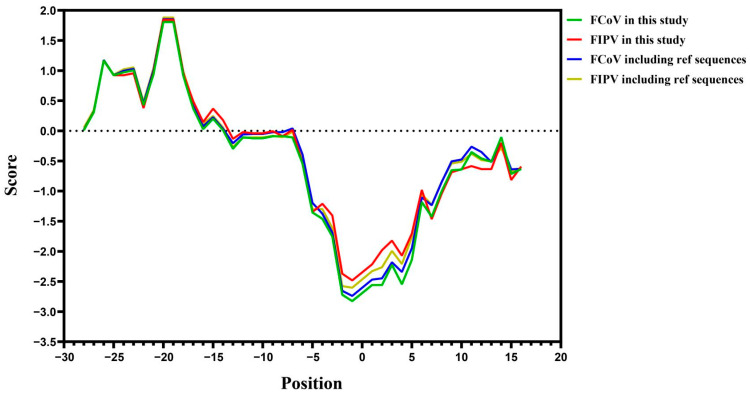
Hydrophobicity of amino acids at S1/S2 cleavage sites. Alignment analyses of the nucleotide sequences of partial S genes between the identified FECV strains and FIPV strains. The furin cleavage site was taken as the zero point, and the protein hydrophobicity value was obtained from the Swiss Institute of Bioinformatics and graphing for hydrophobicity of amino acids at S1/S2 cleavage sites and statistical analysis were performed with GraphPad Prism 8 software (GraphPad Software, 147 Inc., La Jolla, CA, USA). The average value of amino acid hydrophobicity of the obtained 33 non-FIP coronal strains (11 in this study and 22 from GenBank) and 31 FIP strains (7 in this study and 24 from GenBank) from position −28 to position 16 was calculated as the ordinate. The red line and the green line were generated by the data obtained in this study, and the blue and yellow lines were generated including the reference sequence. The referenced strain-related gene bank number is provided in Appendix A.

**Table 1 pathogens-11-00460-t001:** The information of primers.

Prime	Base Sequence	Length	Function	References
F1	GGCAACCCGATGTTTAAAACTGG	223 bp	The detection of FCoV for 3′-UTR	(Herrewegh et al., 1995)
R1	CACTAGATCCAGACGTTAGCTC			
F2	CCACACATACCAAGGCCA	702 bp	FCoV serotyping	(Lin et al., 2009)
R2	CTTAATGCWTWTGTGTCTC			
F3	CCTAGAAAGCCTCAGATGAGTG	Type I: 360 bp Type II: 218 bp		
F4	CAGACCAAACTGGACTGTAC			
R3	CCAAGGCCATTTTACATA			
F5	CAATATTACAATGGCATAATGG	598 bp	FCoV mutation site 23,531 and 23,537 detection	(Chang et al., 2012)
R5	CCCTCGAGTCCCGCAGAAACCATACCTA			
F6	GGCATAATGGTTTTACCTGGTG	142 bp		
R6	TAATTAAGCCTCGCCTGCACTT			
F7	GGCAGAGATGGATCTATTTTTGTTA	1582 bp	Detection of furin cleavage in the S1/S2 site	(Licitra et al., 2013)
R7	ATAATCATCATCAACAGTGCC			
F8	GCACAAGCAGCTGTGATTA	156 bp		
R8	GTAATAGAATTGTGGCAT			

**Table 2 pathogens-11-00460-t002:** Genotyping of FCoV strains.

Clinical States	Total Number of Samples	3′-UTR-Based FCoV Detection	Proportion of S Gene-Based FCoV Detection	FCoV Sequencing and Serotyping		
				Type I	Type II	Both I and II
FIP-suspected cats	81	48/81 (59.3%)	18/81 (22.22%)	18 *	0	0
Non-FIP cats	290	124/290 (42.8%)	67/290 (23.10%)	67 **	0	0
Total number	371	172	85	85	0	0

* Eighteen FIP cats tested positive based on part of the S gene for type I FCoV in this study used for the statistics in Table 3. ** Sixty-seven non-FIP cats tested positive based on part of the S gene for type I FCoV in this study used for the statistics in Table 3.

**Table 3 pathogens-11-00460-t003:** The correlation between serotype and clinical status.

Clinical status	Total Number of Sequences	Type I FCoV	Type II FCoV	χ^2^	*p*	OR	95% CI
	n = 136			0.209	0.648		
FIP cats ^#^	45	42 *	3 **			0.646	0.280–1.494
Non-FIP cats ^##^	91	88 ***	3 ****			1.354	0.603–3.040

^#^ Tissue samples with positive IHC test in suspected FIP cases and positive RT-PCR test in ascites will be regarded as FIP confirmed cases. ^##^ Non-FIP means no FIP symptoms. * Eighteen FIP cats tested positive based on part of the S gene for type I FCoV in this study. Twenty-four FIP isolates were obtained from GenBank for type I FCoV. ** Three FIP cats were obtained from GenBank for type II FCoV. *** Sxty-seven non-FIP cats tested positive based on part of the S gene for type I FCoV in this study. Twenty-one non-FIP cats were obtained from GenBank for type I FCoV. **** Three non-FIP cats were obtained from GenBank for type II FCoV.

**Table 4 pathogens-11-00460-t004:** The correlation between mutation sites 23,531 and 23,537 and clinical status.

Clinical Status	Total Number of Sequences	23,531 or 23,537 Sites Changed	Normal	χ^2^	*p*	OR	95% CI
In this study	n = 85			50.287	0.000		
FIP cats	14	13	1			52	7.301–370.380
Non-FIP cats	71	4	67			0.239	0.101–0.563
Including the reference sequence	n = 138			50.807	0.000		
FIP cats	41	32	9			7.348	3.843–14.051
Non-FIP cats	97	13	84			0.320	0.201–0.508

## Data Availability

The datasets generated during and/or analyzed during the current study can be find in the main text and the Appendix A.

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
