# Peer review of "Epidemiology and Comparative Analyses of the S Gene on Feline Coronavirus in Central China"

_pathogens, 2022, doi:10.3390/pathogens11040460_

Round 1

Reviewer 1 Report

The manuscript entitled “Molecular Epidemiology of Feline Coronavirus on the Basis of Comparative Analyses of the S Gene in Central China” by Hehao Ouyang, Jiahao Liu, Yiya Yin, Shengbo Cao, Rui Yan, Yi Ren, Dengyuan Zhou, Qiuyan Li, Junyi Li, Xueyu Liao, Wanfeng Ji, Bingjie Du, Youhui Si, and Changmin Hu sent for publication to Viral Pathogens, is focusing on very important nowadays topic like Coronavirus associated diseases. The manuscript will be of interest to the scientific community working in that area.  Тhis study provides information on the epidemiology and genetic diversity of the FCoV strains spread in central China. Valuable information for the diagnosis and control of FCoV infections has been prеsented.

However, the authors should address to minor revise the manuscript in order to make it easy to understand. I left my comments and suggestions as follows.

Introduction:

The introduction is well written, the references used are up to date and introduce the main objective.

P35 My recommendation is to include information about Feline Coronavirus distribution not only in China but all over the world.

-P38 The sentence is not fully clear, please give more information about FCoV serotypes. Explain the biotypes in the second sentence.

P44 You are talking about ORFs and proteins, but you don’t explain the genome organization, coding regions, and their proteins products. Please, provide more information about the FCoV proteins and their coding regions.

Materials and methods:

Materials and methods described all protocols.

P112 I would like to see a more detailed explanation of RNA extraction from the feces.

P133 You forgot to mention where the sequencing took place.

Results:

Results are well presented, but I have some suggestions.

I suggest Table 2 be moved to Supplementary materials.  

P202 Put asterisks on the numbers (18 and 67) as in the table, because in the legend below the table suddenly appear numbers that are not in the table and have asterisks.

P227 Increase the size of Figure 1, a not legible inscription of phylogenetic analysis. Indicate the individual colors used in the presentation of the phylogenetic analysis.

P238 Paste Figure 2 immediately after quoting it.

I did not see where you quoted Table 5.

P255 which software did you use to generate the results from Figure 4?

The discussion is well written.

Overall, the manuscript is important for molecular studies of FCoV and its biotypes understanding.

Author Response

Response to Reviewer 1 Comments

Point 1: P35 My recommendation is to include information about Feline Coronavirus distribution not only in China but all over the world.

Response 1: We thank the reviewer for this helpful suggestion. We have included additional information about the prevalence of feline coronaviruses around the world. (manuscript L35 to L36)

Point 2: P38 The sentence is not fully clear, please give more information about FCoV serotypes. Explain the biotypes in the second sentence.

Response 2: Thank the reviewer for this helpful suggestion. We have updated it in main text. (manuscript L39 to L41)

Point 3:P44 You are talking about ORFs and proteins, but you don’t explain the genome organization, coding regions, and their proteins products. Please, provide more information about the FCoV proteins and their coding regions.

Response 3: Thank the reviewer for this helpful suggestion. We have added more information about the FCoV proteins and their coding regions. (manuscript L50 to L58)

Point 4:P112 I would like to see a more detailed explanation of RNA extraction from the feces.

Response 4:

We have added the detailed description of RNA extraction from the feces in the Materials and methods and cited a paper published by Geng et al. as a reference in the manuscript L125 to L131.

Point 5:P133 You forgot to mention where the sequencing took place.

Response 5: Thanks for pointing this out. We have added the information about the sequencing. (manuscript L131 toL132) All the amplified products were sent to AuGCT Biotechnology Co., Ltd for sequencing.

Point 6:I suggest Table 2 be moved to Supplementary materials.

Response 6: Thank the reviewer for this helpful suggestion. We have moved Table 2 to Supplementary materials as Table S2.(Supplementary Information Table S2)

Point 7:P202 Put asterisks on the numbers (18 and 67) as in the table, because in the legend below the table suddenly appear numbers that are not in the table and have asterisks.

Response 7: Thank the reviewer for this helpful suggestion. We put asterisks on the numbers (18 and 67) as in the table.(manuscript L217)

Point 8:P227 Increase the size of Figure 1, a not legible inscription of phylogenetic analysis. Indicate the individual colors used in the presentation of the phylogenetic analysis.

Response 8: Thank the reviewer for this helpful suggestion. Due to the limitation of the article layout, the size of Figure 1 may not be able to be enlarged in the main text. We provide a link to download Figure 1, and readers can view the clear picture locally after downloading according to their needs. Moreover, we supplemented the introduction to the individual colors used in the presentation of the phylogenetic analysis.(manuscript L249 to L267)

Point 9:P238 Paste Figure 2 immediately after quoting it.

Response 9: Thank the reviewer for this helpful suggestion. We modified the position of Figure2 immediately after quoting it.

Point 10:I did not see where you quoted Table 5.

Response 10: Thank the reviewer for check. We quoted Table 5 at the discussion. In the revised manuscript, Table 5 was changed to Table 4.(manuscript L404 to L405)

Point 11:P255 which software did you use to generate the results from Figure 4.

Response 11: Thank the reviewer for this helpful suggestion. The protein hydrophobicity value was obtained from the Swiss Institute of Bioinformatics and graphing for hydrophobicity of amino acids at S1/S2 cleavage sites. The statistical analysis was performed with GraphPad Prism 8 software (GraphPad Software, 147 Inc: La Jolla, CA, USA) We have included the above information in .manuscript L339.

Reviewer 2 Report

Comments for the Author

The manuscript by Ouyang et al describes the occurrence of FIP and the different variants in central China. Therefore, clinical data of 81 suspected FIP cases and 290 other samples from cats were analyzed. In addition to the collection of clinical and cat-specific data, phylogenetic analyses were also performed using the S gene. Furthermore, the influence of the mutation site 23531 and 23537 on the occurrence of FIP was investigated.

Overall, this study is an interesting idea with some untapped potential, even if the results of some analyses have not yet been clearly worked out.

Issues to address:

Comments on the entire manuscript

Many abbreviations are not explained e.g. Table 2 (X2, OR, M), Table 4 (IHC); results 3.4 (T base, A base)

For a general overview, it would be beneficial if Material and Methods already received an overview of which samples and sequences were used for each analysis. From the results it is only insufficiently clear which samples/sequences have contributed to the result

Title

The whole manuscript is more or less an analysis of the S-Gene and less a molecular epidemiology study. Some re-wording is needed.

Table 1

In order to make it easier for the reader to gain an overview, Table 1 should already show which primers and combinations have been used for the various analyses.

Results 3.1

It is difficult to understand why the sample numbers differ so much. It should be pointed out already in the text that the information is not complete and not only as a footnote of Table 2

Table 4

Again, it is unclear how the total number of sequences was arrived at Table 4. In addition, the significance of Table 4 is unclear, since neither the Results nor the Discussion refer to the table 4. It is interesting to see from the data that mainly type I was found in FcoV positive cats and the proportion of type II is low in both FIP positive and negative cats.

Spelling mistake Tyoe I --> Type I

Results 3.3 and Figure 1

The authors mention in 3.3 that 85 FCoV strains belong to type I. In the phylogenetic analysis, however, more partial S genes have obviously been included. Both under 3.3 and in the legend of the figure, it must be made clear what kind of sequences are involved. furthermore, the color scheme should be explained (blue, red gray).

A legend for the different font colors is missing. If the colors have no meaning, black should be used for clarity.

Line 221 It is unclear what “marked with red dots” means. There are no dots in Fig. 1.

Figure 2

It is unclear what the authors conclude from Figure 2. Both part A and part B are poorly described or evaluated in the results and only 2A is mentioned in the discussion. Without any clear conclusions drawn from this Figure it should be removed or moved to the supplemental material.
In addition, the number of sequences mentioned in the text and the figure does not match. In section 3.4, 75 sequences are mentioned with reference to figure 2, whereas 138 sequences are apparently analyzed in the figure.

Figure 3

The set of figures in Figure 3 does not reflect the amount of content that is repeated in all figures. The WebLogo can be omitted since the Frequency Analysis already presents this data. The authors should consider generating only one additional figure besides the amino acid sequence (A/C) to illustrate the results.

In addition, the figures 3C and 3D in particular should be discussed in more detail in 3.5 and the comparison between A/B and C/D should be made clearer.

Figure 4

It is not clear from the text what the authors want to illustrate with Fig. 4. An evaluation and classification of the Hydrophobicity of amino acids at S1/S2 cleavage sites is missing.

Discussion

What conclusions do the authors draw from the results of Table 1 (e.g. clinical status, sex, breed, age and residential density) Part 3.1 of the Results is not discussed.

Line 306-312 Do the authors have any idea why this is? Is there any data on husbandry conditions and the other areas of China mentioned? The section should be discussed in more detail based on existing knowledge.

Author Response

Response to Reviewer 2 Comments

Point 1: Many abbreviations are not explained e.g. Table 2 (X2, OR, M), Table 4 (IHC); results 3.4 (T base, A base)

For a general overview, it would be beneficial if Material and Methods already received an overview of which samples and sequences were used for each analysis. From the results it is only insufficiently clear which samples/sequences have contributed to the result

Response 1: Thank the reviewer for this helpful suggestion.. We have added Tables S1 and S4 to the Supplementary Materials to illustrate the sequences used and their related information. In addition, a comparison table of abbreviations was included in table S5 in the supplementary materials. (Supplementary information L1, L11, L13)

Point 2:

Title

The whole manuscript is more or less an analysis of the S-Gene and less a molecular epidemiology study. Some re-wording is needed.

Response 2: To make it more suitable, we have changed the title to “Epidemiology and Comparative Analyses of the S Gene on Feline Coronavirus in Central China”.(manuscript L2)

Point 3:

Table 1

In order to make it easier for the reader to gain an overview, Table 1 should already show which primers and combinations have been used for the various analyses.

Response 3: Thanks this suggestion. We have listed the analysis of each combination of primers used in Table 1.(manuscript L123)

Point 4:

Results 3.1

It is difficult to understand why the sample numbers differ so much. It should be pointed out already in the text that the information is not complete and not only as a footnote of Table 2.

Response 4:We thank the reviewer for this helpful suggestion. We have added the explanation in main text in the main text L194 to L195. We conducted statistical analysis on specific single information, and did not conduct joint analysis on multiple information, and the data were still statistically significant.

Point 5:

Table 4

Again, it is unclear how the total number of sequences was arrived at Table 4. In addition, the significance of Table 4 is unclear, since neither the Results nor the Discussion refer to the table 4. It is interesting to see from the data that mainly type I was found in FcoV positive cats and the proportion of type II is low in both FIP positive and negative cats.

Response 5:Thanks for this suggestion. We have modified our discussion accordingly (manuscript L224 to L227, L232 to L237 and L380 to L381). We have re-emphasized the significance of the table in the Discussion section and added the specific number of sequences used in the table and the source in a footnote to the table.

Point 7: Spelling mistake Tyoe I --> Type I

Response 7:Thanks for this suggestion. This has been corrected.

Point 8:

Results 3.3 and Figure 1

The authors mention in 3.3 that 85 FCoV strains belong to type I. In the phylogenetic analysis, however, more partial S genes have obviously been included. Both under 3.3 and in the legend of the figure, it must be made clear what kind of sequences are involved. furthermore, the color scheme should be explained (blue, red gray).

A legend for the different font colors is missing. If the colors have no meaning, black should be used for clarity.

Line 221 It is unclear what “marked with red dots” means. There are no dots in Fig. 1.

Response 8: Thanks for this suggestion. We provided information in Table S4 for the reference sequences used in the construction of the phylogenetic tree. And we added the explanation for the color of the scheme and the legend for the different font colors (manuscript L249 to L267). The red dot represented the sequences identified in the present study. 

Point 9:

Figure 2

It is unclear what the authors conclude from Figure 2. Both part A and part B are poorly described or evaluated in the results and only 2A is mentioned in the discussion. Without any clear conclusions drawn from this Figure it should be removed or moved to the supplemental material.

In addition, the number of sequences mentioned in the text and the figure does not match. In section 3.4, 75 sequences are mentioned with reference to figure 2, whereas 138 sequences are apparently analyzed in the figure.

Response 7: Thanks for pointing this out. We have added the corresponding descriptions and conclusions of parts A and B in Figure 2 in the results(manuscript L270 to L275). 138 sequences (75 sequences identified in this study and 63 sequences selected from the GenBank) were analyzed in total. The additional reference sequences are provided in Table S4 of the Supplementary Material to make the number of the sequences more clearly.

Point 10:

Figure 3

The set of figures in Figure 3 does not reflect the amount of content that is repeated in all figures. The WebLogo can be omitted since the Frequency Analysis already presents this data. The authors should consider generating only one additional figure besides the amino acid sequence (A/C) to illustrate the results.

In addition, the figures 3C and 3D in particular should be discussed in more detail in 3.5 and the comparison between A/B and C/D should be made clearer.

Response 10:We thank the reviewer for this helpful suggestion. We have removed the Weblogo in Figure 3 and changed the order of the ABCD sections for better comparative analysis according to the suggestion.

Point 11:

Figure 4

It is not clear from the text what the authors want to illustrate with Fig. 4. An evaluation and classification of the Hydrophobicity of amino acids at S1/S2 cleavage sites is missing.

Response 11:Thanks for this suggestion. We have added more illustration about Fig.4. And the hydrophobicity at S1/S2 is now evaluated in the main text from L315 to L319.

Point 12:

What conclusions do the authors draw from the results of Table 1 (e.g. clinical status, sex, breed, age and residential density) Part 3.1 of the Results is not discussed.

Response 12: Thanks for this suggestion. We have modified our discussion accordingly. (manuscript L367 to L374)

Point 13:

Line 306-312 Do the authors have any idea why this is? Is there any data on husbandry conditions and the other areas of China mentioned? The section should be discussed in more detail based on existing knowledge.

Response 13: At present, there are no other relevant researchs reporting the prevalence of feline coronavirus in central China. Differences in data may be related to the number of samples, the way they were collected and stored, and the specificity and sensitivity of the detection method. The specific reasons for the low prevalence of feline coronavirus in the central region need to be further explored. And we added discussion in the discussion from L361 to L366.

Round 2

Reviewer 2 Report

Dear authors, 

Thank you for the changes that were made during the initial review process. 
However, I still have a few small things that should be improved. 

L34: Blanks are missing between Germany/Malaysia and the brackets

L316-318: Since the weblogo has been removed, this sentence is omitted

Overall, I am pleasantly surprised at how the manuscript has improved. 

Author Response

Response to Reviewer 2 Comments

Point1:Overall, I am pleasantly surprised at how the manuscript has improved.

Response 1: Many thanks to this reviewer for the valuable suggestions that led to a clearer structure and improved presentation of our results. We are also very grateful to the reviewer for his high recognition of our manuscript after revision.

Point2:L34: Blanks are missing between Germany/Malaysia and the brackets.

Response 2: We have added blanks between them.(manuscript L35)

Point3:L316-318: Since the weblogo has been removed, this sentence is omitted.

Response 3: We have modified this section.(manuscript L323)
